# Spatio-temporal pattern and the evolution of the distributional dynamics of county-level agricultural economic resilience in China

Chengmin Li[1], Guoxin Yu[1]*, Haoyu Deng[1], Jian Liu[1], Dongmei Li[2]

1 School of Economics and Management, Xinjiang Agricultural University, Urumqi, China, 2 Business College, Kunming City College, Yunnan, China

* ygx@xjau.edu.cn

## Abstract

Because the complexity of the external environment has put great pressure on the agricultural economy, making it vulnerable, it is necessary to promote a system of resilience in the agricultural economy so that Chinese agriculture can continue to persevere in the face of serious external uncertainties. Therefore, this paper investigates the spatio-temporal pattern and evolution of the distributional dynamics of China's county-level agricultural economic resilience based on 2000–2020 data covering 2,545 counties. The results are as follows: first, from 2000 to 2020, the mean value of China's county-level agricultural economic resilience showed an obvious upward trend, which indicates that China's agricultural economy gradually increased its ability to resist risks and continued to develop in a favourable manner. Specifically, the county-level agricultural economic resilience index of the northeast region grew the most significantly, while the index of county units in the western region was relatively low. Second, the centre of gravity of the spatial distribution of China's agricultural economic resilience gradually migrated to the northwest, showing a dominant direction from northeast to southwest and a tendency to develop from southeast to northwest. Third, the spatial differences in China's agricultural economic resilience generally showed an upward trend, while county-level differences were the main source of the overall differences, followed by inter-provincial differences, inter-municipal differences and inter-regional differences. Additionally, the contribution of county-level differences to the overall differences fluctuated within the range of 54%-58%. Fourth, there is a possibility of localized convergence in China's agricultural economic resilience, which is continuous in spatial effects and has obvious positively correlated spatial effects at different times and in different county spaces.

## Introduction

Currently, China's economy has reached a critical historical point, namely, the shift to high-quality development; this shift has been widely recognized and affirmed [1–3]. Among the many industries that drive high-quality development, agriculture has been one of the most

**Data Availability Statement:** All relevant data are within the paper and its Supporting information file.

**Funding:** This research was funded by the National Natural Science Foundation of China (grant number: 72163032) by Prof. Guoxin Yu, and

Chengmin Li is his doctoral student, and the two belong to the same unit.

**Competing interests:** The authors have declared that no competing interests exist.

prominent driving forces [4, 5], and accelerating the cultivation of its resilience is an important part of building an agricultural powerhouse [6]. COVID-19 was a sudden global shock with significant health, economic and social consequences worldwide, and during this pandemic, the macroeconomic level and agricultural sector were more likely to suffer greater economic vulnerability and greater inequality [7]. In 2021, the United Nations released the "World Economic Forms and Prospects" report, which showed that in 2020, the global economy suffered a severe recession, with the overall size of the economy contracting by 4.3%. Developed economies experienced a more severe recession, contracting by 5.6%, while developing economies contracted by a relatively small 2.5%. For example, the United States economy contracted by 3.5%, and the Japanese economy suffered a recession due to a contraction of 5.3%. In stark contrast, China's economy bucked the trend, with real GDP growth of 2.3% [8–11]. This growth was not only the result of China's aggressive fight against the COVID-19 pandemic but also a reflection of strong economic resilience.

Agriculture is an important part of China's economy, and the resilience of Chinese agriculture is conducive to not only enhancing the international competitiveness of the country's agriculture but also to maintaining the continuity and stability of Chinese economic growth and boosting China's economic development to a new level [12, 13]. In 2021 and 2022, the No. 1 document of the central government noted that we should actively address various domestic and international risks, challenges and difficulties associated with China's agricultural development and ensure stable agricultural production and healthy economic development. In 2023, it was proposed that we should "build a strong agricultural country with a strong supply guarantee, strong scientific and technological equipment, a strong management system, strong industrial resilience, and strong competitiveness". Therefore, in the context of national requirements for the resilience of the agricultural economy, agricultural development must have the capacity to absorb shocks, make adjustments, and to adapt to the changing environment through these adjustments [14]. Unlike measures of financial market resilience, the impact of uncertainty about future outcomes on economic activity is a key driver of the adjustments made by the agricultural sector, with practical implications for the stabilization and resilience of the agricultural economy in the event of a shock [15, 16].

At present, the complexity of the external environment has put great pressure on the agricultural economy, making it vulnerable. It is therefore necessary to promote a system of resilience in the agricultural economy so that Chinese agriculture can continue to persevere in the face of serious external uncertainties and for this resilience to play an obvious role as a "stabilizer" and "ballast" for the steady development of agriculture. Therefore, by analysing the economic resilience of agriculture at different times and at different stages, understanding the characteristics and trends of their dynamic evolution and capturing differences in county-level agricultural economies, it is possible to better respond to major practical problems in modernizing China's agriculture and building a strong agricultural country.

## Literature review

The concept of resilience was proposed by the ecologist Holling in 1973 to evaluate the ability of an ecosystem to sustain and repair itself after a shock [17]. Subsequently, the resilience concept was extended to economics to describe the ability of an economy to recover and expand after a shock. The concept emphasizes the need for economies to be adaptable and flexible to cope with external shocks and to be able to quickly return to normal operations [18–20]. Martin [21] defines economic resilience in terms of four dimensions, i.e., vulnerability, resilience, adaptive capacity and restorative capacity of economic development to shocks. Research on economic resilience started late in China, but some progress has been made in analysing and

evaluating the factors affecting economic resilience and in analysing regional differences and other aspects. Overall, China's economic resilience has shown a fluctuating upward trend, and there are differences in different industries, sectors and regions [22]. Bao and Huang [23] argued that China's economic growth also suffered a certain blow from the exogenous shock of the COVID-19 pandemic and that shadow banking had a positive effect on resolving economic instability, which suggests that finance may also be an important mechanism for influencing the resilience of the agricultural economy. There is also a slow-moving and cyclical component to the development of the agricultural economy, and there is a general economic mechanism underlying this phenomenon. However, few relevant studies have addressed the resilience dimension of the agricultural economy [24]. Thus, Folke emphasized agricultural resilience for the agricultural system to resist external interference and maintain the stability of the original system's ability [25]. The agricultural economy must have the capacity to recover, which is reflected in subjects' ability to reintegrate resources and obtain upgrades in the reverse direction.

In studying economic resilience, the primary issue is the defined measurement methodology. Some studies draw on existing regional economic resilience evaluation models. Early on, Martin [21] analysed the responses of regional economic agents in the face of shocks by empirically measuring these responses along multiple dimensions. Lagravinese [26] then modified the model to study resilience under the impact of an economic crisis in Italy. At present, there are two types of measurements in academia. The first category consists of methods that use a multidimensional indicator system. For example, Briguglio measured macroeconomic development, social development, micro markets and economic governance [27]. Quendler et al. [28] took the Austrian agricultural economy as an example and measured the resilience of Austria between 1995 and 2019 based on six dimensions. In addition, Zhu et al. [29] measured by selecting fiscal revenues and the amount of real foreign investment utilized. The second category is the use of unidimensional indicators of measurement. Researchers use different indicators to measure resilience, but the results are not very comparable; in this case, the employment rate, GDP, etc., are used to explore the resilience of the regional economy [30–33]. However, due to the limitations of the unidimensional indicator method, a more comprehensive and accurate indicator system needs to be explored to better assess and compare the economic resilience of different regions.

Studying the spatio-temporal pattern and dynamic evolution of regional differences in agricultural economic resilience is highly meaningful. Some scholars have analysed the general situation in China and found that the Chinese economy as a whole shows agglomeration. At the same time, however, the regional economy as a whole presents a trend of slow growth [34, 35]. Despite the gradual convergence of the overall economic growth trend, the regional differences that arise are still one of the issues that we need to pay attention to and study [36, 37]. Since the 1980s, China has shown a "balance-imbalance-gradual rebalance" pattern, and its economic centre of gravity has moved to the northwest, experiencing a spatial evolution from the eastern region to the central region [38, 39] and exhibiting a distinctive feature, namely, a tendency towards inertia with reduced volatility [40]. Studies on China's regional economic resilience have mainly focused on the nation as a whole, provinces and regions [41–43], and some cross-studies integrating spatial and agricultural economics have demonstrated that China's agricultural economic resilience as a whole has shown a clear upward trend and regional differences [44]. Economically developed regions, provinces with a larger scale of agricultural production, and the main grain-producing regions are stronger, and economies at the provincial level show a trend of year-by-year optimization [45]. However, most spatial mechanism studies in this literature are limited to regional "adjacency" or "proximity", and there is a lack of correlation thinking from a holistic perspective, with studies mostly being conducted from

the "relationship" perspective rather than the "quantity" perspective. Due to the unstable development of China's agricultural sector in the early years, the volatility of agricultural economic resilience is more obvious, and if the adverse impacts of the subprime mortgage crisis and highly leveraged capital allocation on the market company were reflected in the agricultural sector, this would lead to abnormal volatility in the market, prompting the instability of the agricultural economy to produce risk contagion and spillover effects [46, 47]. Therefore, clarifying the spatial pattern of the resilience of the agricultural economy and its dynamic evolution holds great significance for promoting the high-quality development of the agricultural economy.

In summary, economic resilience is studied in terms of the resistance of economic agents to various internal and external shocks, and research on this topic covers the spatial pattern and dynamic evolution of multiple scales and domains, but few studies target the agricultural sector. Moreover, through the discussion of economic resilience, conducting further research at the county level and understanding the heterogeneous differences between regions hold great significance for realize sustainable economic development and social harmony [48]. As China's economic development model shifts [1–3], it is necessary to pay attention to the resilience of the agricultural economy and to use its leverage effect to solve the dilemmas of agricultural realities, such as shortages of manpower and land, the ageing of farmers, and rising costs [49]. Especially when the economy has been adversely affected, the rural economy has been seriously affected by various measures, although the losses have been reduced. Eliminating the differences between urban and rural areas by constructing a green supply chain system and enhancing the resilience of the agricultural economy holds great significance for China's economic development and social stability [50]. Therefore, in the current "double cycle" development pattern, it is necessary to capture the spatio-temporal pattern and dynamic evolution of regional differences in agricultural economic resilience and to formulate effective policies to promote the high-quality development of the agricultural economy in each region [51].

The contributions of this paper are as follows: first, this article fills a gap in research on the spatio-temporal distributional differences in agricultural economic resilience. At the research scale, the entropy weight method and standard deviation ellipse model are used to understand in depth the regional differences in agricultural economic resilience via data from 2,545 county units. Second, to explore the sources of the spatio-temporal distributional differences in agricultural economic resilience, we reveal the inherent structural characteristics at different scales. Finally, by applying the stochastic kernel density estimation method, the sources of agricultural economic resilience at different stages and the regional differences are investigated at the spatial-geographical level, and the distributional dynamics of China's agricultural economic resilience are examined in detail.

## Research methodology and indicator system construction

### Construction of the indicator system

Academics are still exploring the measurement of agricultural economic resilience and have not yet developed a more unified and authoritative method. Zhang et al. [52] measure agricultural economic resilience through counterfactual labour productivity in agriculture. Ye et al. [53] select indicators based on resistance and reconfiguration. Fully considering the interpretation of resilience in previous studies and data availability, Volkov [14] selects three dimensions: providing income to farm workers, guaranteeing the viability of the farm, and producing affordable food. Žičkienė [54] argues that agricultural economic resilience is a core function of the agricultural sector and should be constituted through three dimensions: providing affordable agricultural products, ensuring farm viability, and creating and sustaining

**Table 1. Index system for rating agricultural economic resilience.**

| Target Level | Indicator Layer | Interpretation of Indicators | Indicator Properties | Indicator Weights |
|---|---|---|---|---|
| Risk resistance | Proportion of the primary industry | Gross primary industry output/GDP | Positive | 0.050 |
| | Rural consumption capacity | Rural retail sales of consumer goods/retail sales of consumer goods | Positive | 0.026 |
| | Per capita economic level of workers in the primary industry | Value added of agriculture, forestry, animal husbandry and fisheries/employees in the primary industry | Positive | 0.021 |
| | Grain output | Total grain production | Positive | 0.110 |
| | Effective irrigation rate | Effective irrigated land area/cultivated land area | Positive | 0.055 |
| | Total mechanical power per unit of arable land area | Total power of agricultural machinery/cultivated land area | Positive | 0.117 |
| | Rural road network accessibility | Mileage of accessible rural roads | Positive | 0.058 |
| | Rural informatization level | Broadband network penetration | Positive | 0.012 |
| Adaptive adjustment | Growth rate of the value added of the primary industry | Growth rate of the value added of the primary industry | Positive | 0.016 |
| | Per capita rural disposable income | Per capita disposable income of rural residents | Positive | 0.055 |
| | Level of consumption expenditure of the rural population | Consumption expenditures of rural residents | Positive | 0.068 |
| Reconstructive creativity | Investment in agricultural fixed assets | Investment in fixed assets of rural households in agriculture, forestry and fisheries | Positive | 0.095 |
| | Level of financial support for agriculture | Amount of fiscal expenditure on agriculture | Positive | 0.042 |
| | Stock of agricultural human capital | Number of students enrolled in higher agricultural institutions at the undergraduate level | Positive | 0.081 |
| | Rural electricity consumption | Electricity consumption by rural residents for productive use | Positive | 0.194 |

decent jobs. The idea of determining resilience indicators based on the pressure, state, and response (PSR) of the evaluated object, as proposed by Martin et al. [30] in the PSR model, has good reference value.

Based on the studies above, this paper combines the connotation of agricultural economic resilience, introduces the PSR index in the PSR model, and measures the level of agricultural economic resilience in terms of three dimensions, namely, risk resistance ability, adaptive adjustment ability and reconstructive creativity ability [55–57]. Risk resistance ability (P) refers to the ability of agriculture to withstand and diversify risks when it suffers shocks. Eight specific characterization indicators are selected, including the proportion of the primary industry, rural consumption capacity, the per capita economic level of employees in the primary industry, grain output, the effective irrigation rate, the total mechanical power per unit of arable land area, the accessibility of the rural road network, and the level of rural informatization. Adaptive adjustment ability (S) refers to the adaptive recovery capacity of the agriculture economy after facing risks and suffering shocks. Three specific indicators are selected, namely, the growth rate of the value added of the primary industry, the per capita disposable income in rural areas, and rural consumer expenditure. reconstructive creativity ability (R) refers to the ability to change and innovate to return to the original state. Four specific indicators are selected, namely, investment in agricultural fixed assets, financial support for agriculture, the stock of agricultural human capital, and rural electricity consumption (see Table 1).

## Research methodology

**Entropy weight method.** Most scholars use hierarchical analysis and the entropy weight method to evaluate indicator systems. The disadvantage of hierarchical analysis is that it cannot avoid the influence of subjective factors, while the entropy weight method only analyses

the number of indicators at all levels, with the aim of avoiding the influence of subjectively assigned factors, and it can assess the comprehensive level of indicators in a more objective way [53, 58]. Therefore, taking into account the overall consideration, this paper adopts the entropy weight method to measure agricultural economic resilience.

The steps for calculating the level of agricultural economic resilience in each district are as follows:

First, the indicators are standardized:

$$y_{ij} = \frac{x_{ij} - x_{i\ min}}{x_{i\ max} - x_{i\ min}} \tag{1}$$

Second, the definition is standardized:

$$Y_{ij} = \frac{y_{ij}}{\sum_{i=1}^{m} y_{ij}} \tag{2}$$

Then, the value of the indicator information entropy and the value of information utility are determined:

$$e_i = -\frac{1}{In_m} \sum_{i=1}^{m} Y_{ij} In_{ij} \tag{3}$$

$$d_i = 1 - e_i \tag{4}$$

where $e$ is information entropy and $d$ is the likelihood of information generation; in general, the smaller the value is, the smaller the degree of disorder, and vice versa.

Finally, the weights of the evaluation indicators are calculated. The information utility value is the largest, indicating that the more important the indicator is, the more important it is to the evaluation. Finally, the weight of the item indicator is obtained as $W_i$:

$$W_i = \frac{d_i}{\sum_{i=1}^{m} d_i} \tag{5}$$

**Standard deviation ellipse model.** This model, which was proposed by Lefever, is a method of using spatial geographic distribution to reveal economic phenomena. It is based on parameters such as the centre of gravity, azimuthal angle θ, and long and short axes of the ellipse to determine the centre of gravity, direction and evolutionary law of the spatio-temporal development of the research object. It reflects the degree of dispersion of the research object in space and determines the direction of distribution of its spatio-temporal development and the trajectory of migration of the centre of gravity [59, 60]:

$$SDE_x = \sqrt{\frac{\sum_{n=1}^{n} (x_i - \bar{X})^2}{n}}, \ \ SDE_y = \sqrt{\frac{\sum_{n=1}^{n} (y_i - \bar{Y})^2}{n}} \tag{6}$$

**Computational modelling of the spatial and temporal differences in agricultural economic resilience.** In 2003, Akita proposed decomposing the Theil index into two stages and at the same time proposed a three-stage nested Theil index model to estimate the degree of spatial variation in different geographic strata [61]. This paper takes the county as the basic spatial unit to expand the sources of spatial differences in the resilience of China's agricultural economy, and it reveals in depth the intrinsic structural characteristics of this resilience at

different levels, such as the regional, inter-provincial, prefectural administrative unit, and county levels [62–64].

i, j, k, and m represent the four levels of regions, namely, the inter-provincial, prefecture-level city (including province-administered counties), and district and county levels, respectively. According to the principle of Theil index decomposition, the decomposition formula of the Theil index within the ijkth group is obtained as follows:

$$T_{ijk} = \sum_i \frac{Y_{ijkm}}{Y_{ijk}} \ln\left(\frac{Y_{ijkm}/Y_{ijk}}{P_{ijkm}/P_{ijk}}\right) \tag{7}$$

where $Y_{ijkm}$ is the level of agricultural economic resilience of m counties in province j and province k in region i; $P_{ijkm}$ is the number of employees in agriculture, forestry, animal husbandry and fisheries in province j, province k, and city or county m in region i; and $T_{ijk}$ is the intra-group disparity in prefecture-level cities. According to Eq (7), we can define $T_{ijk}$ as a measure of the degree of difference.

Then, the decomposition of the Theil index within the ijth group is obtained further as follows:

$$T_{ij} = \sum_k \frac{Y_{ijk}}{Y_{ij}} T_{ijk} + \sum_i \frac{Y_{ijk}}{Y_{ij}} \ln\left(\frac{Y_{ijk}/Y_{ij}}{P_{ijk}/P_{ij}}\right) \tag{8}$$

where $T_{ij}$ is decomposed into within-group and between-group disparities in prefecture-level municipalities, which is defined as a measure of the degree of disparity in the agricultural economic resilience of the prefecture-level municipalities according to Eq (8).

Furthermore, the Theil index within group i is decomposed as follows:

$$T_i = \sum_j \frac{Y_{ij}}{Y_i} T_{ij} + \sum_i \frac{Y_{ij}}{Y_i} \ln\left(\frac{Y_{ij}/Y_i}{P_{ij}/P_i}\right) \tag{9}$$

where the decomposition of $T_i$ into intra- and inter-provincial Thayer indices is defined as a measure of the degree of inter-provincial agricultural economic resilience variation according to Eq (9).

Finally, the decomposition of the Theil index within group i is obtained as follows:

$$T = \sum_i \frac{Y_i}{Y} T_i + \sum_i \frac{Y_i}{Y} \ln\left(\frac{Y_i/Y}{P_i/P}\right) \tag{10}$$

where T is the total Theil index.

In regard to nesting equations, by substituting Eqs (2)–(4) into Eq (5) level by level, a three-stage nested Theil exponential decomposition is obtained as follows:

$$T = \sum_i\sum_j\sum_l\sum_m \frac{Y_{ijkm}}{Y_{ijk}} \ln\left(\frac{Y_{ijkm}/Y_{ijk}}{P_{ijkm}/P_{ijk}}\right) + \sum_i\sum_j\sum_l \frac{Y_{ijk}}{Y_{ij}} \ln\left(\frac{Y_{ijk}/Y_{ij}}{P_{ijk}/P_{ij}}\right)$$
$$+ \sum_i\sum_j \frac{Y_{ij}}{Y_i} \ln\left(\frac{Y_{ij}/Y_i}{P_{ij}/P_i}\right) + \sum_i \frac{Y_i}{Y} \ln\left(\frac{Y_i/Y}{P_i/P}\right) \tag{11}$$

$$T = T_{Inter-regional} + T_{Inter-provincial} + T_{Inter-City} + T_{In-City} \tag{12}$$

**Computational modelling of the dynamic evolution of agricultural economic resilience.** Stochastic kernel density estimation is an improved version of the traditional Markov chain approach. It obtains probability density estimates and three-dimensional dynamic distribution maps of agricultural economic resilience through the kernel density function, and it is used to describe the distributional pattern of agricultural economic resilience under different conditions [65]. This analysis is based on the results obtained from the bivariate kernel estimation imported into the Gaussian kernel function for calculation, and the expression is as follows:

$$g(y|x) = \frac{f(x, y)}{f(x)} \tag{13}$$

In Eq (8), f(x) denotes the marginal kernel density function of x, and f(x,y) is the joint kernel density function of x and y. The expression is as follows:

$$f(x, y) = \frac{1}{Nh_x h_y} \sum_{i=1}^{N} K_x\left(\frac{X_i - x}{h_x}\right) K_y\left(\frac{Y_i - x}{h_y}\right) \tag{14}$$

Eq (9) is the bivariate kernel estimation expression, where N is the number of observations, $h_x$ and $h_y$ are the bandwidths, K denotes the normal kernel function, the distributions of $X_i$ and x denote the observed and changed values of agricultural economic resilience in a county in year t, respectively, and $Y_i$ and y are the observed and changed values of agricultural economic resilience in a county in year t+n, respectively.

## Sample data

**Regional division.** According to the regional division office of the National Bureau of Statistics of China and the regional division approach of Cui and other scholars [66], 2,545 counties in 31 provinces (regions and cities) of China (excluding Hong Kong, Macao, and Taiwan) are divided into four major regions: 625 counties in the eastern region, 639 counties in the central region, 1,056 counties in the western region, and 225 counties in the northeast region.

**Data sources.** The agricultural economic resilience data used in this paper are obtained from the 2000–2020 China County Statistical Yearbook, China Regional Economic Statistical Yearbook, and China Urban Statistical Yearbook, as well as statistical yearbook data on provinces (districts and municipalities), prefectural-level cities, and counties. Additionally, the county panel data are obtained by collecting and organizing the relevant yearbooks. Then, missing values in some county agricultural economic resilience data are filled in through interpolation. Finally, balanced panel data covering 2,545 Chinese counties used in this paper are obtained.

# Measurement and spatio-temporal patterns of agricultural economic resilience in China

## Measurement and characteristics of the spatial distribution of agricultural economic resilience in China

The entropy weight method was used to calculate the agricultural economic resilience index scores of 2,545 counties in China, as shown in Table 2. A larger value of this index indicates a higher degree of agricultural economic resilience in that provincial or municipal node, and vice versa. Due to the large number of county units, this paper presents the county agricultural economic resilience index by weighted average at the provincial level.

**Table 2.  Trends in the evolution of agricultural economic resilience in China, 2000–2020.**

| Provinces | 2000 | 2005 | 2010 | 2015 | 2020 | Average |
|---|---|---|---|---|---|---|
| Beijing | 0.123 | 0.142 | 0.176 | 0.206 | 0.232 | 0.177 |
| Tianjin | 0.072 | 0.085 | 0.120 | 0.143 | 0.142 | 0.115 |
| Hebei | 0.299 | 0.305 | 0.337 | 0.365 | 0.367 | 0.335 |
| Shanxi | 0.142 | 0.161 | 0.188 | 0.200 | 0.209 | 0.181 |
| Neimenggu | 0.188 | 0.219 | 0.255 | 0.289 | 0.318 | 0.258 |
| Liaoning | 0.174 | 0.192 | 0.225 | 0.236 | 0.237 | 0.214 |
| Jilin | 0.142 | 0.159 | 0.177 | 0.196 | 0.212 | 0.181 |
| Heilongjiang | 0.196 | 0.223 | 0.266 | 0.292 | 0.331 | 0.274 |
| Shanghai | 0.116 | 0.129 | 0.155 | 0.184 | 0.211 | 0.162 |
| Jiangsu | 0.333 | 0.325 | 0.389 | 0.437 | 0.464 | 0.389 |
| Zhejiang | 0.227 | 0.230 | 0.291 | 0.335 | 0.378 | 0.291 |
| Anhui | 0.214 | 0.234 | 0.268 | 0.230 | 0.340 | 0.277 |
| Fujian | 0.125 | 0.144 | 0.184 | 0.213 | 0.181 | 0.181 |
| Jiangxi | 0.154 | 0.179 | 0.197 | 0.220 | 0.255 | 0.204 |
| Shandong | 0.425 | 0.420 | 0.470 | 0.512 | 0.508 | 0.465 |
| Henan | 0.374 | 0.389 | 0.425 | 0.454 | 0.497 | 0.429 |
| Hubei | 0.205 | 0.237 | 0.271 | 0.301 | 0.347 | 0.276 |
| Hunan | 0.204 | 0.232 | 0.264 | 0.300 | 0.342 | 0.272 |
| Guangdong | 0.236 | 0.242 | 0.297 | 0.346 | 0.445 | 0.312 |
| Guangxi | 0.143 | 0.165 | 0.188 | 0.212 | 0.250 | 0.195 |
| Hainan | 0.047 | 0.057 | 0.075 | 0.088 | 0.108 | 0.076 |
| Chongqing | 0.095 | 0.107 | 0.125 | 0.145 | 0.169 | 0.130 |
| Sichuan | 0.261 | 0.274 | 0.321 | 0.360 | 0.414 | 0.328 |
| Guizhou | 0.104 | 0.116 | 0.156 | 0.192 | 0.229 | 0.162 |
| Yunnan | 0.150 | 0.178 | 0.219 | 0.246 | 0.273 | 0.217 |
| Xizang | 0.030 | 0.037 | 0.048 | 0.059 | 0.083 | 0.052 |
| Shanxi | 0.163 | 0.190 | 0.205 | 0.222 | 0.246 | 0.208 |
| Gansu | 0.121 | 0.137 | 0.155 | 0.176 | 0.195 | 0.158 |
| Qinghai | 0.039 | 0.047 | 0.062 | 0.073 | 0.088 | 0.063 |
| Ningxia | 0.056 | 0.066 | 0.077 | 0.091 | 0.105 | 0.080 |
| Xinjiang | 0.106 | 0.131 | 0.171 | 0.203 | 0.237 | 0.173 |
| Average | 0.170 | 0.186 | 0.218 | 0.245 | 0.273 | 0.220 |
| Eastern Region | 0.200 | 0.208 | 0.249 | 0.283 | 0.309 | 0.250 |
| Central Region | 0.215 | 0.240 | 0.269 | 0.295 | 0.332 | 0.273 |
| Western Region | 0.121 | 0.139 | 0.165 | 0.189 | 0.217 | 0.169 |
| Northeast Region | 0.171 | 0.191 | 0.223 | 0.241 | 0.260 | 0.223 |

According to the results of the measurements and combined with Fig 1, in the past two decades, the agricultural economic resilience of each province in China has shown an upward trend. This finding indicates that China has achieved a significant enhancement in agricultural economic resilience and that the trend of sustained positive development is obvious. In particular, in two specific provinces, Shandong and Henan, the mean values of agricultural economic resilience are higher, reaching 0.465 and 0.429, respectively. As large agricultural provinces, Shandong and Henan have continuously optimized their supply structures, built regional brands, and worked hard to develop agricultural demonstration zones in recent years, which has further led to the realization of an innovative, green, and efficient modernized agricultural system. However, the mean values for Qinghai and Xizang are lower, 0.063 and 0.052,

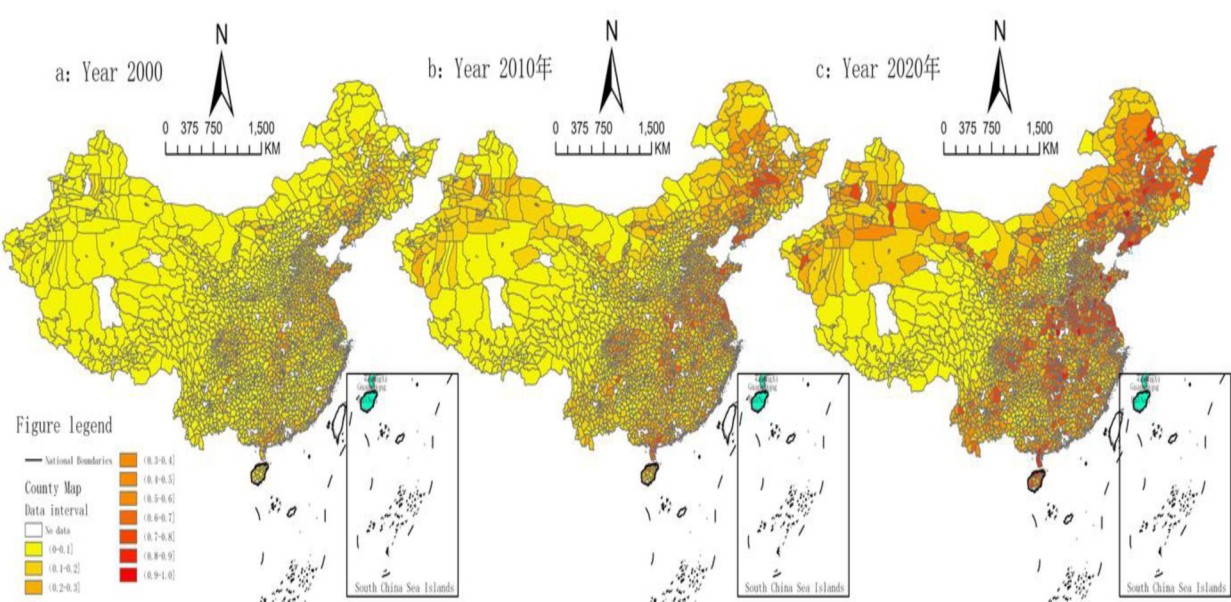

**Fig 1. Spatial distribution of agricultural economic resilience in Chinese counties.** Map Review Number for the Chinese: GS (2022) 1873 (Supervised by the Department of Natural Resources), without modification of the base map. Notes: Excluding data on Hong Kong, Macao and Taiwan.

respectively. The main reason is that the level of economic development in these regions is relatively low, and a lack of economic, technological and human resources obstructs agricultural economic resilience improvement. The analysis above also directly reflects the fact that the level of development of agricultural economic resilience is uneven across regions in China, with some regions lagging behind others. In addition, although Beijing, Tianjin and Shanghai have strong comprehensive economic strength, their agricultural economic resilience is low, and they rank relatively low in the country. This finding is mainly due to the nature of these cities, with Beijing being the political and cultural centre of China, Shanghai being the economic centre of China, and Tianjin being the economic centre of the Bohai Rim region. The non-agricultural industry has a strong foundation, and the contribution of agriculture to the local economy is limited. Additionally, due to the high land prices and goods prices in these areas, the number of agricultural employees continue to decline. This trend poses a threat to the sustainability of agriculture.

As shown in Fig 1, for the agricultural economic resilience of the top ten and bottom ten counties, China's agricultural economic resilience index was generally low in 2000, and the level of agricultural economic resilience in most county units was within the range of (0–0.162]. With the evolution of spatio-temporal dynamics as well as the continuous improvement in the level of agricultural development, 2010 and 2020 still showed an imbalance in regional agricultural economic resilience, the absolute differences between regions continued to increase, and the imbalanced characteristics in the spatial geography became increasingly obvious. Specifically, comparing the evolution of agricultural economic resilience in 2000, 2010 and 2020, we see that the level of county-level agricultural economic resilience in the northeast region was relatively high, in which the county-level agricultural economic resilience indices of Liaoning Province and Heilongjiang Province exhibited the most obvious growth, followed by Henan, Hubei, Hunan and other provinces in the central region and Shandong, Jiangsu, Guangdong and other provinces in the eastern region. The economic resilience index increased significantly, and the level of agricultural economic resilience in most counties

exceeded 0.3 in 2020, with Zhuanghe city in Liaoning Province and Jianli County in Hubei Province having the highest level of agricultural economic resilience within the county units nationwide. Finally, the level in the western region was lower. Specifically, the index of county units in Tibet, Ningxia and other provinces was lower than 0.1, making the western region the region with the lowest level of agricultural economic resilience. The level of other provinces in the western region was in the range of (0.3–0.5], converging to a level with counties in other regions with lower levels of agricultural economic resilience.

## Shift in the centre of gravity and standard deviation ellipse analysis of agricultural economic resilience in Chinese counties

Based on calculations, five characteristic time points were selected—2000, 2005, 2010, 2015 and 2020—and the centre of gravity standard deviation ellipse model was applied to generate standard deviation ellipses of regions in different periods and to obtain the migration trajectories, trends in distribution, and other parameters of the centre of gravity to analyse the migration path of agricultural economic resilience.

As shown in Fig 2, in terms of the distribution of the centre of gravity, the centre of gravity of the five characteristic time points varies within the range of (114°21'41.2"~114°10'25.4"E, 32°51'20.3"~33°35'31.0"N), which implies that agricultural economic resilience is generally higher in southeastern counties and is lower in the western and northern regions. Based on the migration trajectory of the centre of gravity, from 2000 to 2005, the centre of gravity moved north-westward from Runan County to near the junction of Runan County and Suiping County, followed by a north-westward movement from Runan County to Xiping County from 2005 to 2010. From 2010 to 2015, it gradually shifted northwards to Luohe city, and finally, it

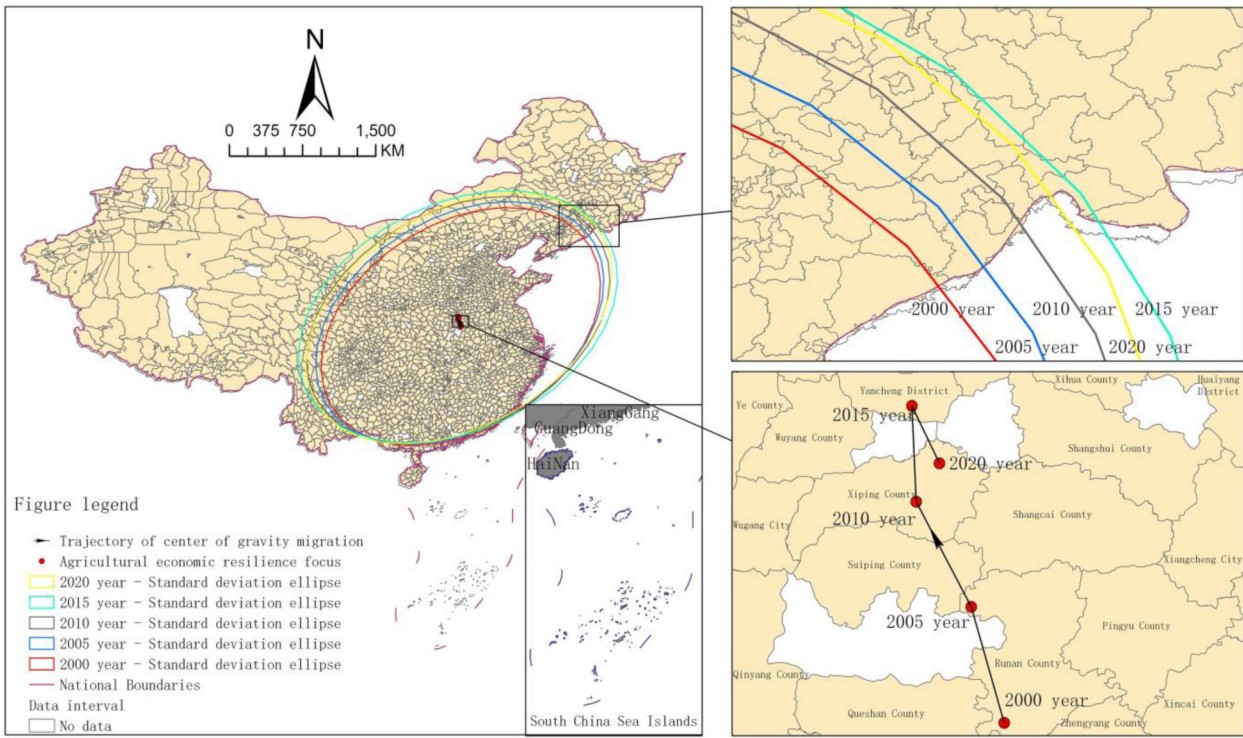

**Fig 2. Transfer paths of agricultural economic resilience in China.** Map Review Number for the Chinese: GS (2022) 1873 (Supervised by the Department of Natural Resources), without modification of the base map. Notes: Excluding data on Hong Kong, Macao and Taiwan.

shifted back to Xiping County in the southeast direction from 2015 to 2020. Overall, the centre of gravity of China's county-level agricultural economic resilience distribution shifted the greatest distance in the northwest direction, indicating that counties in western China were higher than the national average in terms of the spatio-temporal dynamic evolution from 2000 to 2020.

According to the standard deviation ellipse distribution, the development of China's agricultural economic resilience from 2000 to 2020 basically exhibited a spatial distribution pattern from northeast to southwest. The area of the ellipse shows a gradual increasing trend, from 412.44 km$^2$ in 2000 to 503.85 km$^2$ in 2015, followed by a decrease to 471.78 km$^2$ in 2020. These findings indicate that there was a spatial dispersion trend in China's county agricultural economic resilience. The change trend of the azimuthal angle θ showed an upward trend in the range of 59˚09'57.3–60˚52'59.3 from 2000 to 2020, and the largest change occurred in 2015, reaching 64˚54'37.5". These findings indicate that the directionality of China's county agricultural economic resilience dispersion to the northwest might become increasingly obvious. In terms of semi-axis length, the lengths of the long and short semi-axes increased from 14 km to 15 km and from 9 km to 10 km from 2000 to 2020, respectively. These findings are similar to the trends of the elliptical area and azimuthal angle θ, which also showed an increasing trend. The highest value was in 2015, indicating that the agricultural economic resilience of Chinese counties had a trend dominated by the direction from northeast to southwest and the direction from southeast to northwest.

## Spatial differences in agricultural economic resilience in China and their decomposition

Based on the results of the three-stage nested Theil index decomposition, the inherent structural characteristics of the differences in agricultural economic resilience in China's counties were revealed in depth at different levels, such as the regional, inter-provincial, prefecture-level administrative unit, and county levels. According to Fig 3, the overall variation in China's

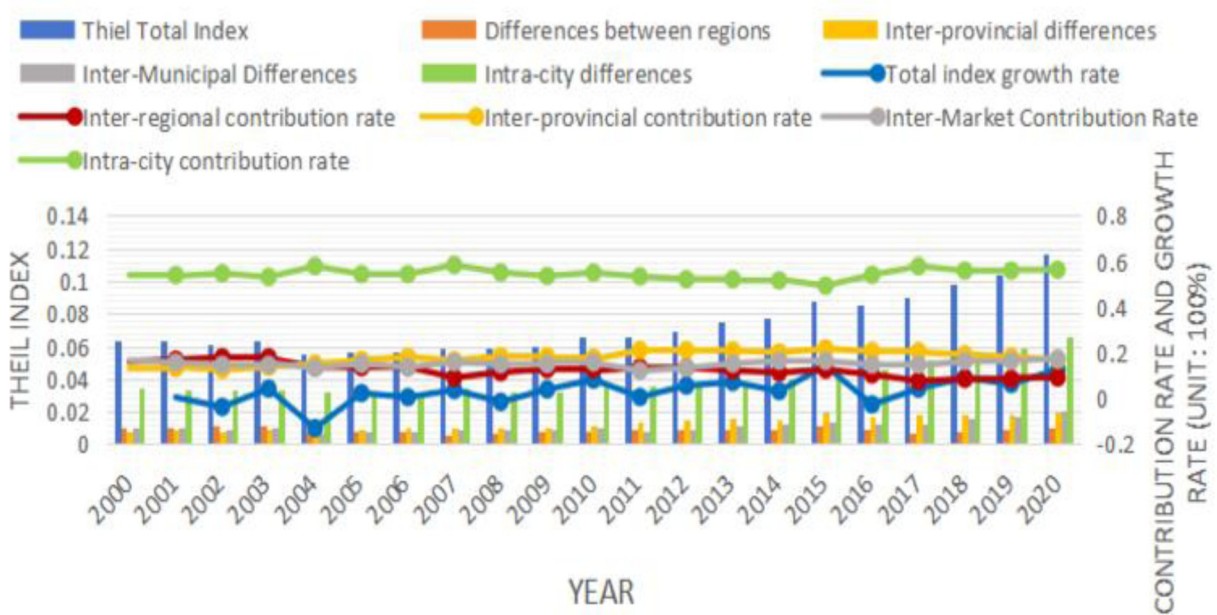

**Fig 3. Three-stage nested Theil exponential decomposition results.** Notes: Excluding data on Hong Kong, Macao and Taiwan.

county-level agricultural economic resilience increased, from 0.063 in 2000 to 0.117 in 2020. According to the specific analysis of the results, first, at the county level, which exhibited the greatest variation, the intra-city (county) variation fluctuated in the range of 0.03–0.06, with an upward trend from 2000 to 2020. This variation was the most significant source and contributed the most to the overall variation, fluctuating in the range of 54%-58%, followed by the inter-provincial level. The inter-provincial variation increased from 0.008 to 0.02 from 2000 to 2020, showing a slow upward trend, and its contribution to the overall variation increased from 13% to approximately 16%, reaching as high as 21% in 2015, making it the second largest source of overall variation. The inter-provincial level is followed by the prefecture-level city level. The variation among cities first decreased from 0.01 in 2000 to 0.007 in 2004 and then gently rose to 0.2 in 2020. Finally, at the regional level, inter-regional differences declined from 0.01 in 2000 to 0.006 in 2020, showing a slow downward trend, and these differences the fourth largest source of overall differences. Thus, the differences in county-level agricultural economic resilience in China are a fundamental source of the spatial differences in China's overall agricultural economic resilience.

To further examine the spatial pattern of agricultural economic resilience variation in China's counties, we used spatial geography software to visualize the variation in county-level agricultural economic resilience. The spatial distribution in Fig 4 shows that the differences in northeast China are generally large. These differences include Zhuanghe city, Changwu County, the Dawa District, and Kaiyuan city in Liaoning Province; Nong'an County in Jilin Province; and Fuyuan city, Nengjiang County, Zhao Dong city, Mohe city, and Tahe County in Heilongjiang Province. Most of the differences in agricultural economic resilience in these counties are in the range of [0.00082–0.001781] and are generally higher than the national average. Most of the differences between the central and eastern regions are in the range of [0.000253–0.000820] and are generally in the middle level of the country. In contrast, the differences in county-level agricultural economic resilience in western China are generally small, especially in the cities of Xinjiang, Xizang, and Gansu and in other western inland provinces.

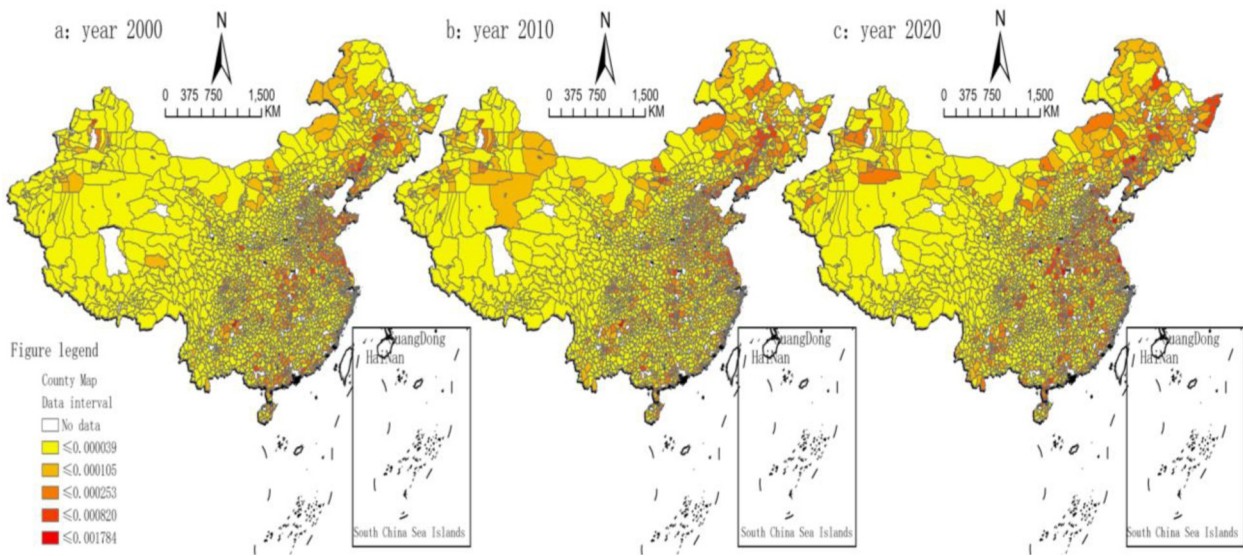

**Fig 4. Trends in the evolution of differences in agricultural economic resilience in China, 2000–2020.** Map Review Number for the Chinese: GS (2022) 1873 (Supervised by the Department of Natural Resources), without modification of the base map. Notes: Excluding data on Hong Kong, Macao and Taiwan.

The differences in county-level agricultural economic resilience are mostly within the interval of [0.000039–0.000105] and are generally lower than the national average. These findings indicate that the resilience of China's county-level agricultural economies shows some differences. That is, the resilience of county-level agricultural economies in the eastern region is relatively high, while that in the western region is relatively low. Furthermore, this difference has been relatively stable in terms of its spatial distribution over the past 20 years.

## Evolution of the distributional dynamics of agricultural economic resilience in China's counties

### Unconditional kernel density estimation of agricultural economic resilience in China's counties

In Fig 5A and 5D, the X-axis and Y-axis represent the agricultural economic resilience levels in years t and t+3, respectively, and the Z-axis represents the kernel density. Specifically, the agricultural economic resilience of Chinese counties mainly shows two trends: when the level of agricultural economic resilience is lower than 0.25, the kernel density contour is concentrated on the 45˚ diagonal, indicating that the distribution in year t+3 is the same as that in year t. When the level is higher than 0.3, the kernel density contour appears above the 45˚ diagonal, indicating that the county shifted from a low level to a high level between years t and t+3 and showing an upward trend. Analysing the distributional direction of the contours, we find that China's agricultural economic resilience may locally converge in the interval of (0.4–0.6]. That is, there is a possibility of shifting from being a low-level county to being a high-level county, while high-level counties are more stable. These findings validate the analysis in Table 1 above that western counties have relatively low levels and tend to stabilize at certain low to medium levels. The implication is that agricultural economic development is lagging behind in the western region, which faces greater difficulties and challenges. In contrast, the agricultural economic resilience of some counties in the northeast, central and eastern regions shows an upward trend. This finding suggests that these regions have become more resilient to risks and have made some degree of progress.

### Stochastic kernel density estimation of the spatial conditions of agricultural economic resilience in China's counties

In Fig 5B and 5E, the X-axis and Y-axis are the level of agricultural economic resilience in geographically neighbouring counties and the focal county, respectively, and the Z-axis indicates the kernel density. Analysing the overall dynamics of the probability density distribution, we find that the kernel density contours mainly show two major clusters. If the agricultural economic resilience levels are defined as low or high with 0.2 as the threshold, then when the agricultural economic resilience levels of geographically adjacent counties are less than 0.2 in year t, the kernel density contours are concentrated on the 45˚ diagonal, revealing a low-low clustering phenomenon, At this stage, there is a significant positive spatial correlation. When the level of geographically adjacent counties in year t is less than 0.2, the kernel density contour is concentrated in the position below the 45˚ diagonal, revealing the phenomenon of high-low clustering. At this stage, the overall agricultural economic resilience of geographically adjacent counties in year t is at a low level of spatial effect with respect to that of the focal county in year t. This finding verifies the analysis in Fig 1 above that the agricultural economy in Chinese counties generally lacks resilience and is characterized by spatial convergence. In addition, counties that geographically neighbour the focal county have a significant impact. That is, if a county has a high level of agricultural economic resilience with its geographic neighbours,

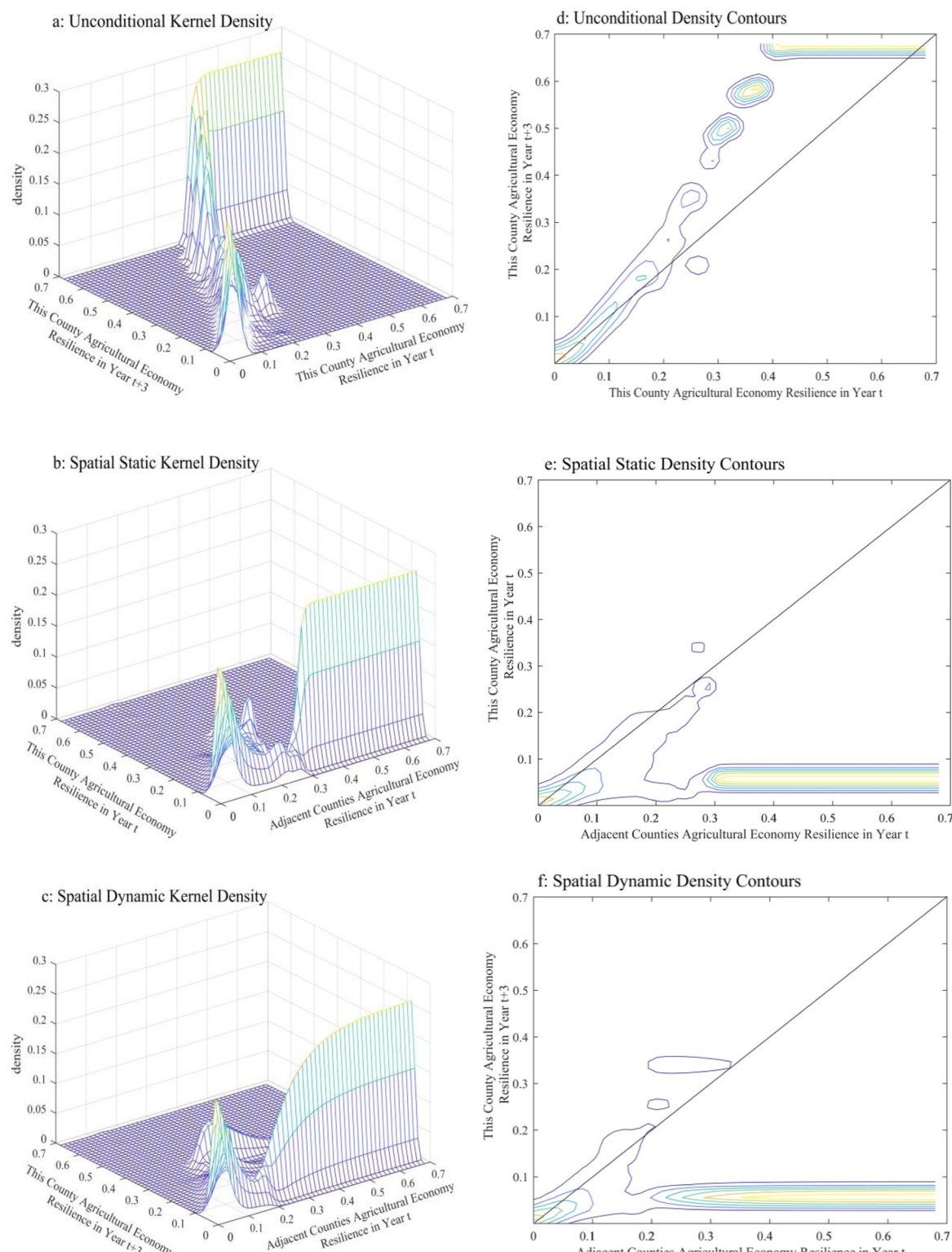

**Fig 5. Kernel density and density contour of the agricultural economic resilience of counties in China.** Notes: Excluding data on Hong Kong, Macao and Taiwan.

then the county is also relatively high, and vice versa. This positive spatial effect suggests that the interrelationships between geographically neighbouring counties are to some degree important for agricultural economic resilience. Therefore, when formulating agricultural development policies, the impacts of geographically neighbouring counties on the focal county should be taken into account to improve the overall resilience of the agricultural economy.

## Dynamic kernel density estimation of the spatial conditions of agricultural economic growth in China

In Fig 5C and 5F, the X-axis and Y-axis show the level of agricultural economic resilience of geographically neighbouring counties in year t and the focal county in year t+3, respectively, and the Z-axis represents the kernel density. This section utilizes the stochastic kernel estimation method to further reveal whether the agricultural economic resilience of Chinese counties has a spatial dynamic effect. Specifically, the dynamic Kernel estimation results under the spatial condition are basically consistent with the stochastic estimation Kernel estimation results, and we find that, even when considering the time lag and spatial lag conditions, geographically neighbouring counties in year t still have a significant positive impact on the focal county in year t and in year t+3 under the spatial lag condition. That is, this spatial effect has persistence, showing a clear, positive spatial correlation effect.

## Discussion

### Conclusions

Using the standard deviation ellipse, the three-stage nested Theil index and stochastic kernel density estimation, this paper investigates the spatio-temporal pattern and evolution of the distributional dynamics of the resilience of China's agricultural economy based on 2000–2020 data covering 2,545 counties. The results show that (1) in the spatial pattern of China's agricultural economic resilience, the mean value of the agricultural economic resilience of China's provinces showed an obvious upward trend from 2000 to 2020. This finding indicates that the risk resistance ability of China's agricultural economy gradually increased and that the trend of sustained good development is obvious. Specifically, the county agricultural economic resilience index for the northeast region had the most obvious growth, while county units in the western region had a lower level of agricultural economic resilience. (2) The centre of gravity of the spatial distribution of China's agricultural economic resilience gradually migrated to the northwest, showing a dominant direction from northeast to southwest and a tendency to develop in the direction from northwest from southeast. Moreover, the spatial differences in China's agricultural economic resilience generally show an upward trend, while the county-level differences in China's agricultural economic resilience are the main source of the overall differences, followed by inter-provincial differences, inter-municipal differences and inter-regional differences. (3) In the study of the distributional dynamics of China's agricultural economic resilience, the level of agricultural economic resilience in China under unconditional kernel density estimation shows a growing trend and the possibility of local convergence. Under spatial conditional kernel density estimation, the spatial effect of the resilience of China's county-level agricultural economy shows persistence, and there is an obvious positive spatial correlation effect at different times and in different county spaces. These findings suggest that China's agricultural economic resilience and the county spatial pattern will have an effect on each other in the future and that China's agricultural economic resilience will increase with increasing inter-area differences.

## Policy recommendations

All of the results above, especially the latter results, have important policy implications: (1) The government should promote the development of the agricultural productive service industry and help the level of agricultural economic resilience rise rapidly. According to the results of the agricultural economic resilience index, the difference between the agricultural economic resilience of some counties and that of large agricultural provinces is large, mainly due to the low level of agricultural technology and the relative lack of scientific and technological talent, which impedes agricultural economic resilience improvement. Therefore, by relying on policies such as the State Council's Guiding Opinions on Accelerating the Development of the Productive Service Industry to Promote the Adjustment and Upgrading of the Industrial Structure (Guo Fa [2014] No. 26), the government should increase its support for the agricultural productive service industry, develop pre-production, mid-production and post-production agricultural productive services for agricultural production in an orderly manner, and strive to improve the level of services in logistics, information, finance and marketing. The transformation and upgrading of the agricultural industrial structure should be promoted to enhance the risk resistance ability in regions with lower agricultural economic resilience. (2) The construction of agricultural regional cooperation mechanisms should be promoted, and the level of agricultural economic resilience should be enhanced. From the perspective of the spatial distribution of agricultural economic resilience in China's counties, the spatial effects of agricultural economic resilience are persistent and affect each other. Therefore, we can draw on the effectiveness of the implementation of the Yangtze River Delta Regional Ecological Environment Protection Plan, the Action Plan for the Continuous Improvement of Air Quality, and the Chengdu-Chongqing Twin Cities Economic Circle Construction Plan. According to the regional spatial distribution pattern of the agricultural economy, we can classify city clusters that have a high level of agricultural economic resilience or a high concentration of agricultural economic resilience and promote agricultural resilience improvement in regions with low agricultural economic resilience by constructing an agricultural economic resilience regional collaboration mechanism. (3) The government should increase its support for the central and western regions to improve their agricultural risk resistance ability. It should also increase its support for agricultural development in the central and western regions to eliminate the imbalance in the level of regional agricultural economic resilience. According to the policy advice of the Guidance on Financial Support for Comprehensively Promoting Rural Revitalization and Accelerating the Construction of a Strong Agricultural Country, we should focus on supporting central and western provinces with obvious spillover effects of agricultural economic resilience, strengthening the construction of basic agricultural capacity and supporting conditions in these regions, attaching more importance to improving the comprehensive agricultural production capacity of these regions, and advancing the transformation and upgrading of agriculture as well as high-quality development. At the same time, rural areas have been taken as the main battlefield for the development of social work and the integration of urban and rural elements. Rural areas have played an important role in the two-way flow of multiple factors, setting up special earmarked funds and projects, and increasing the volume of agricultural capital and other livelihood funds to consolidate the foundation of agricultural and rural development and thus enhance the risk resistance ability of agriculture in areas of low agricultural economic resilience in the central and western regions.

## Limitations and future recommendations

There are limitations to our findings. First, the agricultural economic resilience indicator system needs to be expanded. Based on data reliability and availability, this paper analyses the

construction of an agricultural economic resilience indicator system from the level of resilience to risk resistance ability, adaptive adjustment ability, and reconstructive creativity ability, and the innovativeness of agricultural economic resilience research in China is not deep enough. Subsequent research can include agricultural financial mechanisms for exploration, such as agricultural insurance and government subsidies [23, 50]. Second, the driving factors of agricultural economic resilience need to be screened. This paper only summarized the previous literature, selecting risk resistance ability, adaptive adjustment ability and reconstructive creativity ability as well as other aspects of measurement. However, agricultural economic resilience is a relatively complex concept, and the degree of resilience tends to be affected by a variety of factors, which are not included in this paper due to length limitations. Future research can pay further attention to the natural conditions and humanistic characteristics of agricultural economic resilience as driving factors of economic resilience improvement. Third, the mechanism of inter-regional agricultural resilience and the internal and external environments need to be studied in depth. This paper analyses the agricultural economic resilience system of 2,545 counties in China; however, the decomposition and exploration of the inter-regional indicator system are insufficient, and we expect to conduct an in-depth exploration of the mechanism of synergistic development between the internal and external environments of regions and agricultural resilience in the future. In summary, our indicator system and conclusions may not be applicable when considering agricultural economic resilience in all countries and regions of the world.

In future research, a broader sample of countries and regions could be analysed, including the resilience of the agricultural economy in the Asian region or globally, as well as the spillover effects of China on the resilience of the agricultural economy in neighbouring regions (e.g., Central Asia) or neighbouring countries. Another avenue for future research may be to apply agricultural economic resilience measurement methods, spillover effects, and drivers to economic activities, and further research should focus on exploring the paths of agricultural economic resilience improvement based on finer-grained data to provide more insights and recommendations for high-quality agricultural development.

## Supporting information

**S1 Data.**
(XLSX)

## Acknowledgments

The authors are grateful for the patient review and helpful suggestions from the editor of this journal, as well as the anonymous reviewers.

## Author Contributions

**Conceptualization:** Chengmin Li, Guoxin Yu, Dongmei Li.

**Data curation:** Chengmin Li, Jian Liu, Dongmei Li.

**Formal analysis:** Haoyu Deng, Jian Liu, Dongmei Li.

**Funding acquisition:** Guoxin Yu.

**Investigation:** Chengmin Li, Haoyu Deng.

**Methodology:** Chengmin Li, Guoxin Yu, Jian Liu.

**Project administration:** Guoxin Yu.

**Resources:** Chengmin Li, Guoxin Yu, Haoyu Deng, Jian Liu, Dongmei Li.

**Software:** Haoyu Deng, Dongmei Li.

**Supervision:** Guoxin Yu.

**Validation:** Jian Liu.

**Visualization:** Chengmin Li, Jian Liu.

**Writing – original draft:** Chengmin Li.

**Writing – review & editing:** Guoxin Yu.

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
