## [Editor Report · Decision Letter 0]

18 Oct 2023

PONE-D-23-33108Spatiotemporal Pattern and Distribution Dynamic Evolution in County Agricultural Economy Resilience in ChinaPLOS ONE

Dear Dr. YU,

Thank you for submitting your manuscript to PLOS ONE. After careful consideration, we feel that it has merit but does not fully meet PLOS ONE’s publication criteria as it currently stands. Therefore, we invite you to submit a revised version of the manuscript that addresses the points raised during the review process.

We look forward to receiving your revised manuscript.

Kind regards,

Difang Huang, Ph.D.

Academic Editor

PLOS ONE

Journal Requirements:

4. We note that Figures 1, 2 and 4  in your submission contain map/satellite images which may be copyrighted. All PLOS content is published under the Creative Commons Attribution License (CC BY 4.0), which means that the manuscript, images, and Supporting Information files will be freely available online, and any third party is permitted to access, download, copy, distribute, and use these materials in any way, even commercially, with proper attribution. For these reasons, we cannot publish previously copyrighted maps or satellite images created using proprietary data, such as Google software (Google Maps, Street View, and Earth). For more information, see our copyright guidelines: http://journals.plos.org/plosone/s/licenses-and-copyright.

a. You may seek permission from the original copyright holder of Figures 1, 2 and 4 to publish the content specifically under the CC BY 4.0 license.  

Additional Editor Comments:

1. Improving the literature review:

We recommend that you incorporate some relevant papers from the provided publication list into your literature review. These papers can help strengthen the context of your study and provide valuable insights into the factors affecting agricultural economic resilience. Specifically, we suggest the following papers:

- Bao, Z., & Huang, D. (2021). Shadow banking in a crisis: Evidence from FinTech during COVID-19. Journal of Financial and Quantitative Analysis, 56(7), 2320–2355.

- Chen, M., Li, N., Zheng, L., Huang, D., & Wu, B. (2022). Dynamic correlation of market connectivity, risk spillover and abnormal volatility in stock price. Physica A: Statistical Mechanics and Its Applications, 587, 126506.

- Li, N., Chen, M., Gao, H., Huang, D., & Yang, X. (2023). Impact of lockdown and government subsidies on rural households at early COVID-19 pandemic in China. China Agricultural Economic Review, 15(1), 109–133.

Li, N., Chen, M., & Huang, D. (2022). How Do Logistics Disruptions Affect Rural Households? Evidence from COVID-19 in China. Sustainability, 15(1), 465.

Wu, B., Huang, D., & Chen, M. (2023). Estimating contagion mechanism in global equity market with time-zone effect. Financial Management, 52, 543–572.

Yu, D., & Huang, D. (2023a). Cross-sectional uncertainty and expected stock returns. Journal of Empirical Finance, 72, 321–340.

Yu, D., & Huang, D. (2023b). Option-Implied Idiosyncratic Skewness and Expected Returns: Mind the Long Run. Available at SSRN 4323748.

Yu, D., Huang, D., & Chen, L. (2023). Stock return predictability and cyclical movements in valuation ratios. Journal of Empirical Finance, 72, 36–53.

The paper by Bao and Huang (2021) investigates the role of shadow banking during the COVID-19 crisis, which can provide insights into the financial mechanisms that may affect agricultural economic resilience. Chen et al. (2022) explore the dynamic correlation of market connectivity, risk spillover, and abnormal volatility in stock prices, which can help understand the potential impact of market fluctuations on agricultural economic resilience. Li et al. (2023) examine the impact of lockdown and government subsidies on rural households during the early COVID-19 pandemic in China, which can offer valuable information on the effects of external shocks and policy interventions on agricultural economic resilience.

By incorporating these papers into your literature review, you can provide a more comprehensive understanding of the factors affecting agricultural economic resilience and better contextualize your study within the existing literature.

2. Detailed comments to improve the submission:

- Please provide a more detailed explanation of the methodology used to calculate the agricultural economic resilience index. This will help readers better understand the process and the rationale behind the chosen method.

- Consider discussing the potential limitations of your study, such as data availability, the representativeness of the sample, and the generalizability of the findings to other regions or countries.

- It would be helpful to include a discussion on the policy implications of your findings. For example, what specific policies or interventions can be implemented to improve agricultural economic resilience in the regions with lower resilience levels?

---

## [Author Response · Author response to Decision Letter 0]

9 Jan 2024

Dear Difang Huang Editors and Reviewers:

Thank you for your letter and for the reviewers’ comments concerning our manuscript entitled “Spatio-temporal Pattern and the evolution of the distributional dynamics of county-level agricultural economic resilience in China” (ID: PONE-D-23-33108). Those comments are all valuable and very helpful for revising and improving our paper, as well as the important guiding significance to our research. We have studied comments carefully and have made correction which we hope meet with approval.

---

## [Decision Letter · Decision Letter 1]

25 Jan 2024

PONE-D-23-33108R1Spatio-temporal pattern and the evolution of the distributional dynamics of county-level agricultural economic resilience in ChinaPLOS ONE

Dear Dr. YU,

Thank you for submitting your manuscript to PLOS ONE. After careful consideration, we feel that it has merit but does not fully meet PLOS ONE’s publication criteria as it currently stands. Therefore, we invite you to submit a revised version of the manuscript that addresses the points raised during the review process.

**Reviewer 1 **
**Comments:**

At the current stage of China's high-quality economic development, the driving factors in the agricultural sector make a significant contribution. Using the entropy weight method based on data from 2000 to 2020, covering 2545 counties, this study calculated the economic elasticity of 180 indicators in agriculture. The research focused on the spatio-temporal patterns and evolutionary dynamics of China's county-level agricultural economic resilience. The findings hold crucial reference significance for agricultural development.

Revisions are as follows:

1. Please indicate the map review number for the Chinese map in the article, and ensure consistency in elements like the legend and compass orientation.

2. Tables should be formatted using the three-line table format.

3. For the discussion section, consider organizing paragraphs to present one point per paragraph for a more aesthetically pleasing layout.

4. While the introduction introduces the research background, there is limited mention of the summary and shortcomings of past research. Additionally, the innovative aspects and practical significance of this study are not clearly emphasized. It is recommended to supplement this information.

**Reviewer 2 **
**Comments:**

1. The article should first write about the construction of the indicator system, and then write about the research methods

2. The research methods are all mature and there is no need to introduce them in such detail

3. There are certain problems with the indicator system, such as the lack of necessary connection between rural electricity consumption and innovation capacity

4. The sample range is said to be 31 provinces, but there are no 31 provinces in Table 2

5. The paper lacks innovation

We look forward to receiving your revised manuscript.

Kind regards,

Fuyou Guo, (Ph.D.

Academic Editor

PLOS ONE

Journal Requirements:

Additional Editor Comments:

Reviewer 1 Comments:

At the current stage of China's high-quality economic development, the driving factors in the agricultural sector make a significant contribution. Using the entropy weight method based on data from 2000 to 2020, covering 2545 counties, this study calculated the economic elasticity of 180 indicators in agriculture. The research focused on the spatio-temporal patterns and evolutionary dynamics of China's county-level agricultural economic resilience. The findings hold crucial reference significance for agricultural development.

Revisions are as follows:

1. Please indicate the map review number for the Chinese map in the article, and ensure consistency in elements like the legend and compass orientation.

2. Tables should be formatted using the three-line table format.

3. For the discussion section, consider organizing paragraphs to present one point per paragraph for a more aesthetically pleasing layout.

4. While the introduction introduces the research background, there is limited mention of the summary and shortcomings of past research. Additionally, the innovative aspects and practical significance of this study are not clearly emphasized. It is recommended to supplement this information.

Reviewer 2 Comments:

1. The article should first write about the construction of the indicator system, and then write about the research methods

2. The research methods are all mature and there is no need to introduce them in such detail

3. There are certain problems with the indicator system, such as the lack of necessary connection between rural electricity consumption and innovation capacity

4. The sample range is said to be 31 provinces, but there are no 31 provinces in Table 2

5. The paper lacks innovation

Reviewers' comments:

Reviewer's Responses to Questions

**Comments to the Author**

1. If the authors have adequately addressed your comments raised in a previous round of review and you feel that this manuscript is now acceptable for publication, you may indicate that here to bypass the “Comments to the Author” section, enter your conflict of interest statement in the “Confidential to Editor” section, and submit your "Accept" recommendation.

Reviewer #1: All comments have been addressed

Reviewer #2: (No Response)

2. Is the manuscript technically sound, and do the data support the conclusions?

Reviewer #1: Yes

Reviewer #2: Partly

3. Has the statistical analysis been performed appropriately and rigorously? 

Reviewer #1: Yes

Reviewer #2: Yes

4. Have the authors made all data underlying the findings in their manuscript fully available?

Reviewer #1: Yes

Reviewer #2: No

5. Is the manuscript presented in an intelligible fashion and written in standard English?

Reviewer #1: Yes

Reviewer #2: Yes

6. Review Comments to the Author

Reviewer #1: At the current stage of China's high-quality economic development, the driving factors in the agricultural sector make a significant contribution. Using the entropy weight method based on data from 2000 to 2020, covering 2545 counties, this study calculated the economic elasticity of 180 indicators in agriculture. The research focused on the spatio-temporal patterns and evolutionary dynamics of China's county-level agricultural economic resilience. The findings hold crucial reference significance for agricultural development.

Revisions are as follows:

1. Please indicate the map review number for the Chinese map in the article, and ensure consistency in elements like the legend and compass orientation.

2. Tables should be formatted using the three-line table format.

3. For the discussion section, consider organizing paragraphs to present one point per paragraph for a more aesthetically pleasing layout.

4. While the introduction introduces the research background, there is limited mention of the summary and shortcomings of past research. Additionally, the innovative aspects and practical significance of this study are not clearly emphasized. It is recommended to supplement this information.

Reviewer #2: 1. The article should first write about the construction of the indicator system, and then write about the research methods

2. The research methods are all mature and there is no need to introduce them in such detail

3. There are certain problems with the indicator system, such as the lack of necessary connection between rural electricity consumption and innovation capacity

4. The sample range is said to be 31 provinces, but there are no 31 provinces in Table 2

5. The paper lacks innovation

7. PLOS authors have the option to publish the peer review history of their article (what does this mean?). If published, this will include your full peer review and any attached files.

Reviewer #1: No

Reviewer #2: No

---

## [Author Response · Author response to Decision Letter 1]

27 Jan 2024

Dear Reviewer:

Thank you for your letter and for the reviewers’ comments concerning our manuscript entitled “Spatio-temporal pattern and the evolution of the distributional dynamics of county-level agricultural economic resilience in China” (ID: PONE-D-23-33108R1). Those comments are all valuable and very helpful for revising and improving our paper, as well as the important guiding significance to our research. We have studied comments carefully and have made correction which we hope meet with approval, the detailed corrections are listed below:

1.SUGGESTIONS 1 FROM REVIEWER 1: Please indicate the map review number for the Chinese map in the article, and ensure consistency in elements like the legend and compass orientation.

We respond to the comments from the reviewer 1：Thank you very much for your suggestion. First of all, for the issue of review number, we have replaced "Review No." with "Map Review Number for the Chinese:" under Figs. 1, 2, 4, etc., in which GS (2022) 1873 is the map review number. Secondly, regarding the compasses and legends of Figs. 1 and 4, we have improved the compass of Fig. 2. Moreover, in our article, the legends of Figures 1 and 4 are consistent with the compass, whereas the legend of Figure 2 is somewhat different because of what it represents. Therefore, we changed the compass and did not modify the legend. Finally, if there are still problems with our charts, we will continue to improve them. Thank you for your valuable comments and we hope that our changes will satisfy you.

2.SUGGESTIONS 2 FROM REVIEWER 1: Tables should be formatted using the three-line table format.

We respond to the comments from the reviewer 1：We are very sorry that the formatting error led to such problems in our article. According to the reviewer's suggestion, we have revised Table 1 and 2 to three-line table format, and we are very grateful to the reviewer for his valuable comments, and we hope that our revisions will be satisfactory to you.

3.SUGGESTIONS 3 FROM REVIEWER 1: For the discussion section, consider organizing paragraphs to present one point per paragraph for a more aesthetically pleasing layout.

We respond to the comments from the reviewer 1：Thank you very much for your suggestions and giving us the opportunity to improve the article. We have added three sub-chapters to the “Discussions” section of the article: conclusions, policy recommendations, limitations and future recommendations. We hope that our revisions will satisfy you.

4.SUGGESTIONS 4 FROM REVIEWER 1: While the introduction introduces the research background, there is limited mention of the summary and shortcomings of past research. Additionally, the innovative aspects and practical significance of this study are not clearly emphasized. It is recommended to supplement this information.

We respond to the comments from the reviewer 1：The reviewer's suggestion was very relevant, pointing out that our article was missing a part of the content. We explained that we have included the summary of past studies and their shortcomings in the second part of the article (Literature Review). We also point out that most of the spatial mechanism studies in the literature are limited to regional "proximity" or "similarity", but lack the overall perspective of correlation, and are mostly based on "relationship" rather than "quantity". We have not yet considered the correlation from a holistic perspective, and we have mostly studied the spatial and temporal patterns of agricultural economic resilience from the perspective of "relationship" rather than "quantity". Meanwhile, in the last paragraph, we propose three possible contributions or innovations of this study. We hope that our explanations will satisfy the reviewers, and we thank them for their suggestions.

6.SUGGESTIONS 1 FROM REVIEWER 2: The article should first write about the construction of the indicator system, and then write about the research methods.

We respond to the comments from the reviewer 2：We would like to thank the reviewers for their valuable comments, which helped us to improve our article. According to the reviewer's suggestion, we have moved "construction of the indicator system" to the front of "research methods". We hope that you will be satisfied with our revision.

7.SUGGESTIONS 2 FROM REVIEWER 2: The research methods are all mature and there is no need to introduce them in such detail.

We respond to the comments from the reviewer 2：The suggestions made by the reviewers were very reasonable. Since the reviewers suggested a detailed explanation of the methodology for measuring the economic resilience of agriculture during the first round of review, we have introduced the entropy right method extensively. Therefore, we are also very grateful for this valuable advice and have abridged the introduction of the entropy right method. As for the other research methods, we think there is no main content missing and a short introduction has been made, so no addition or deletion of content has been made, and we have only revised and improved the entropy right method. We thank the reviewers for their suggestions, and hope that our revision will satisfy you.

8.SUGGESTIONS 3 FROM REVIEWER 2: There are certain problems with the indicator system, such as the lack of necessary connection between rural electricity consumption and innovation capacity.

We respond to the comments from the reviewer 2：Many thanks to the reviewers for pointing out the problems with the indicators in our article, such as the lack of the necessary link between rural electricity consumption and innovation capacity. Here we make explanations:

1. electricity is an important driver of innovation: electricity is the basis of modern agriculture and scientific and technological development, and it provides the necessary energy for innovative activities. With a stable supply of electricity, all kinds of innovative activities in rural areas can be carried out smoothly, from the research and development of agricultural technology to the operation of rural enterprises, all of which need the support of electricity.

2. Rural electricity consumption reflects the level of economic development: Generally speaking, the electricity consumption of a region can be used as an important indicator to measure its level of economic development. Higher electricity consumption usually means that there is more business and industrial activity in the area, which provides the necessary economic base and market environment for innovation.

3. The relationship between electricity consumption and education, training and access to information, e.g.: in rural areas, an increase in the availability of electricity may also mean an increase in education and training opportunities. More electricity means more possibilities to use modern technological means (e.g. the Internet) to access knowledge and information, which is crucial for fostering a sense of creativity and enhancing innovation.

4. Electricity consumption has an important relationship with technology adoption and innovation: an increase in rural electricity consumption is likely to promote the adoption and diffusion of agricultural technologies, including more efficient irrigation systems, agricultural mechanization, etc. This not only directly enhances agricultural production, but also increases the availability of agricultural technologies. This not only directly improves the efficiency of agricultural production, but also provides the necessary material basis for agricultural innovation.

5. Rural electricity consumption is related to infrastructure development: Government investment in infrastructure in rural areas, including the construction of electricity networks, may directly contribute to rural economic development and innovative activities. For example, by providing electricity, the government may create a more innovation-friendly environment in rural areas.

In summary, rural electricity consumption not only directly reflects the level of economic development and commercial and industrial activity in the region, but also indirectly affects innovation capacity by influencing education and access to information, technology adoption, and the building of an innovative environment. Therefore, rural electricity consumption is an important reference point when assessing the indicator of reconfigured innovation capacity.

Finally, we would like to thank the reviewers for their valuable comments and hope that our explanations will satisfy you.

9.SUGGESTIONS 4 FROM REVIEWER 2: The sample range is said to be 31 provinces, but there are no 31 provinces in Table 2.

We respond to the comments from the reviewer 2：Many thanks to the reviewers for their suggestions, we rechecked the contents of Table 2 and determined that it was a range that included 31 provinces. It is possible that we did not have a three-line table, which caused the table to be unclear, so we have changed the form of the table to a three-line table. We hope that our explanation is satisfactory to you.

10.SUGGESTIONS 5 FROM REVIEWER 2: The paper lacks innovation.

We respond to the comments from the reviewer 2：Many thanks to the reviewers for pointing out the lack of innovation in our article. In this regard, we have done the following two aspects to improve the innovation: on the one hand, most scholars' research scale is only centered on provincial and prefecture-level city units [39-47], but our innovation in this study is to take 2,545 county units as the scope of the study, which improves the innovation of the study in terms of the research scale. On the other hand, we apply the methods of "three-stage nested Theil index" and "stochastic Kernel density estimation" to the spatial differences and dynamic distribution of agricultural economic resilience, which can serve as a reference for related studies. Moreover, these two methods have certain novelty in themselves. To sum up, although there are many existing studies on the toughness of agricultural economy, there is still a big difference with our study, and the purpose of this study is to analyze the shortcomings of China's agricultural economic development, in order to build a strong agricultural country, and to improve the toughness of agricultural economy in the lower counties, which will help to improve China's agricultural economy as a whole. Finally, we are very grateful to the reviewers for pointing out the problems, and we hope that our explanations will satisfy the reviewers. Meanwhile, we are already studying the mechanism of digital finance, production services, digital economy and other elements on the resilience of the agricultural economy, which is used to make up for the current shortcomings in front of us.

---

## [Decision Letter · Decision Letter 2]

1 Mar 2024

Spatio-temporal pattern and the evolution of the distributional dynamics of county-level agricultural economic resilience in China

PONE-D-23-33108R2

Dear Dr. Yu,

We’re pleased to inform you that your manuscript has been judged scientifically suitable for publication and will be formally accepted for publication once it meets all outstanding technical requirements.

Kind regards,

Fuyou Guo, (Ph.D.

Academic Editor

PLOS ONE

Additional Editor Comments (optional):

Reviewers' comments:

Reviewer's Responses to Questions

**Comments to the Author**

1. If the authors have adequately addressed your comments raised in a previous round of review and you feel that this manuscript is now acceptable for publication, you may indicate that here to bypass the “Comments to the Author” section, enter your conflict of interest statement in the “Confidential to Editor” section, and submit your "Accept" recommendation.

Reviewer #1: All comments have been addressed

2. Is the manuscript technically sound, and do the data support the conclusions?

Reviewer #1: Yes

3. Has the statistical analysis been performed appropriately and rigorously? 

Reviewer #1: Yes

4. Have the authors made all data underlying the findings in their manuscript fully available?

Reviewer #1: Yes

5. Is the manuscript presented in an intelligible fashion and written in standard English?

Reviewer #1: Yes

6. Review Comments to the Author

Reviewer #1: The relevant comments and opinions have been revised and improved, and the paper is accepted for publication.

7. PLOS authors have the option to publish the peer review history of their article (what does this mean?). If published, this will include your full peer review and any attached files.

Reviewer #1: No

---

## [Editor Report · Acceptance letter]

27 Mar 2024

PONE-D-23-33108R2 

PLOS ONE

Dear Dr. Yu, 

I'm pleased to inform you that your manuscript has been deemed suitable for publication in PLOS ONE. Congratulations! Your manuscript is now being handed over to our production team.

Kind regards, 

on behalf of

Associate professor Fuyou Guo 

Academic Editor

PLOS ONE